# BLACK-BOX APPROXIMATION AND OPTIMIZATION WITH HIERARCHICAL TUCKER DECOMPOSITION

## ABSTRACT

We develop a new method HTBB for the multidimensional black-box approximation and gradient-free optimization, which is based on the low-rank hierarchical Tucker decomposition with the use of the MaxVol indices selection procedure. Numerical experiments for 14 complex model problems demonstrate the robustness of the proposed method for dimensions up to 1000, while it shows significantly more accurate results than classical gradient-free optimization methods, as well as approximation and optimization methods based on the popular tensor train decomposition, which represents a simpler case of a tensor network.

## 1 INTRODUCTION

Many physical and engineering models can be represented as a real function (output), which depends on a multidimensional argument (input) and looks like

$$y = \mathsf{f}(\boldsymbol{x}) \in \mathbb{R}, \quad \boldsymbol{x} = [x_1, x_2, \ldots, x_d]^T \in \Omega \subset \mathbb{R}^d. \tag{1}$$

Such functions often have the form of a black-box (BB), *i. e.*, the internal structure and smoothness properties of f remain unknown. Its discretization on some multi-dimensional grid results in a multidimensional array (tensor[1]) $\mathcal{Y} \in \mathbb{R}^{N_1 \times N_2 \times \ldots \times N_d}$ that collects all possible discrete values of the function (1) inside the domain $\Omega$, i.e.,

$$\mathcal{Y}[n_1, n_2, \ldots, n_d] = \mathsf{f}(x_1^{(n_1)}, x_2^{(n_2)}, \ldots, x_d^{(n_d)}). \tag{2}$$

Storing such a tensor often requires too much computational effort, and for large values of the dimension $d$, this is completely impossible due to the so-called curse of dimensionality (the memory for storing data and the complexity of working with it grows exponentially in $d$). To overcome it, various compression formats for multidimensional tensors are proposed: Canonical Polyadic decomposition aka CANDECOMP/PARAFAC (CPD) Harshman et al. (1970), Tucker decomposition Tucker (1966), Tensor Train (TT) decomposition Oseledets (2011), Hierarchical Tucker (HT) decomposition Hackbusch & Kühn (2009); Ballani et al. (2013), and their various modifications. These approaches make it possible to approximately represent the tensor in a compact low-rank (i.e., low-parameter) format and then operate with the compressed tensor.

The TT-decomposition is one of the most common compression formats Cichocki et al. (2016; 2017). There is an algebra for tensors in the TT-format (i.e., TT-tensors): we can directly add and multiply TT tensors, truncate TT tensors (reduce the so-called TT-rank, i. e, the number of storage parameters), integrate and contract TT tensors. It is important that effective algorithms have been developed Kapushev et al. (2020); Ahmadi-Asl et al. (2021); Chertkov et al. (2023b) for approximating BB like (1) and (2) in the TT-format, that is, for constructing an approximation (surrogate model) using only a small number of explicitly computed BB values. In recent years, new efficient algorithms have also been proposed Sozykin et al. (2022); Nikitin et al. (2022); Chertkov et al. (2023a) for the second important problem associated with gradient-free optimization of such

---

[1]By tensors we mean multidimensional arrays with a number of dimensions $d$ ($d \geq 1$). A two-way tensor ($d = 2$) is a matrix, and when $d = 1$ it is a vector. For scalars we use normal font, we denote vectors with bold letters and we use upper case calligraphic letters ($\mathcal{A}, \mathcal{B}, \mathcal{C}, \ldots$) for tensors with $d > 2$. The $(n_1, n_2, \ldots, n_d)$th entry of a $d$-way tensor $\mathcal{Y} \in \mathbb{R}^{N_1 \times N_2 \times \ldots \times N_d}$ is denoted by $\mathcal{Y}[n_1, n_2, \ldots, n_d]$, where $n_k = 1, 2, \ldots, N_k$ ($k = 1, 2, \ldots, d$) and $N_k$ is a size of the $k$-th mode. Mode-$k$ slice of such tensor is denoted by $\mathcal{Y}[n_1, \ldots, n_{k-1}, :, n_{k+1}, \ldots, n_d] \in \mathbb{R}^{N_k}$.

BB, that is, finding an approximate minimum or maximum value based only on queries to the BB. Such TT-based methods of surrogate modeling (in particular, the TT-cross algorithm Oseledets & Tyrtyshnikov (2010)) and gradient-free optimization (in particular, the TTOpt algorithm Sozykin et al. (2022)) have shown their effectiveness for various multidimensional problems, including compression and acceleration of neural networks, data processing, modeling of physical systems, etc.

However, the TT-decomposition is one of the simplest special cases of a tensor network: it is a linear network or a degenerate tree, and it has a number of limitations related to weak expressiveness and instability for the case of significantly large dimensions. The HT-format is potentially more expressive and robust Buczyńska et al. (2015); thus, it makes it possible to approximate more complex functions with fewer parameters. Taking into account TT-Cross and TTOpt algorithms which use the well-known MaxVol approach Goreinov et al. (2010); Mikhalev & Oseledets (2018), in this work we develop new methods of surrogate modeling and gradient-free optimization based on the HT-format, and our main contributions are the following:

- we develop a new black-box approximation method HT-cross based on the HT-decomposition and the rectangular MaxVol index selection procedure;

- we develop a new gradient-free optimization method HTOpt based on the HT-decomposition and the rectangular MaxVol index selection procedure;

- we implement the proposed HT-cross and HTOpt algorithm as a unified method HTBB (Hierarchical Tucker for Black-Box) for surrogate modeling and optimization of multidimensional functions given in the form of a black-box and share it as a publicly available python package;[2]

- we apply our approach HTBB to 14 different complex model functions with input dimensions up to 1000 and demonstrate its significant advantage in the accuracy and robustness for the same budget in comparison with the TT-cross method for approximation and with the TTOpt, and classical gradient-free SPSA and PSO methods for optimization problems.

## 2 MOTIVATION AND OVERALL IDEA

HT-format (see Fig. 3 for a visual example of HT structure) is more expressive and robust Buczyńska et al. (2015) than simpler forms of tensor networks (for example, the well-known TT-decomposition), which makes it potentially possible to apply it for complex functions. Thus, it seems important to develop new approximation and optimization methods based on it. We are inspired by a simpler, but carefully designed TT-format and implement analogues of the known methods TT-cross and TTOpt on its basis for the HT-decomposition. The TT-cross algorithm Oseledets & Tyrtyshnikov (2010) adaptively calls the BB and iteratively builds the TT-surrogate until a given accuracy is reached or the BB access budget is exhausted. During this construction, the so-called Maximum Volume submatrix search (MaxVol) procedure Goreinov et al. (2010) is used to find a close to the dominant matrix of the tensor unfolding. As mentioned in the cited paper, the MaxVol algorithm is closely related to finding the maximum element in a given matrix — the submatrix, obtained by MaxVol, contains values close to the maximum modulus values of the tensor. This effect can be used to find the quasi-maximal element in the tensor, and the corresponding algorithm is called TTOpt Sozykin et al. (2022).

Thus, our goal is to extend the above algorithms to the HT tree structure. To do this, we need to solve several algorithmic problems related to the fact that the two-dimensional HT structure has ambiguities that are not present in the one-dimensional TT structure. In particular, it is necessary to determine a) the sequence to traverse the HT tree (for TT structure the traversal is done sequentially from left to right and back); and b) how to form a matrix $A$, which is the input to the MaxVol algorithm (in TT cores there is a dedicated index, thanks to which it is unfolded, and we know exactly which indices are row and which are column indices of the matrix $A$. In HT structure cores are more symmetric and the choice of indices for rows and columns of the matrix $A$ to which MaxVol is applied is not obvious).

Mathematically, we solve two problems using slight modifications of the same algorithm. The first problem is the approximation of a given discrete black box, *i.e.*, a function $f(X) \in \mathbb{R}$ whose

---

[2]The program code with the proposed approach and numerical examples, given in this work, is publicly available in the repository ANONYMIZED.

arguments are a set of natural numbers $I \in \{1, \ldots, n_1\} \times \{1, \ldots, n_2\} \times \cdots \times \{1, \ldots, n_d\}$, by means of a low-parameter HT representation. As a result, we have an HT representation $\mathcal{H}^3$ such that by some norm

$$\|f - \mathcal{H}\| < \epsilon$$

for small $\epsilon$ which for the given function $f$ may depend on the HT-ranks and the number of calls $N$ to the function $f$. In this setting, we adaptively (at runtime of the algorithm, based on previous values of the function) access the function $f$ values, thus justifying the name "black box". The second problem is to find the extreme (minimum or maximum) value of the discrete black box $f$ described above. In other words, for no more than a given number of accesses (budget) $N$ we want to obtain such a value of the argument $I_0$ that

$$|f(I_0) - f(I_{\text{true}})| < \epsilon,$$

where $I_{\text{true}}$ is the exact value of the minimum or maximum (may not be unique). In such a setting, we also adaptively call the black box $f$. Note that this formulation does not assume the existence of any low-parameter representation, but, as in the case of the TTOpt algorithm, we expect that if $f$ is well approximated by some HT representation with a given accuracy, then the results of our algorithm will be better.

Both of these problems are solved by iteratively updating the set of arguments. This update takes into account the hierarchical structure of HT cores and takes place on the basis of the obtained function values on a special combination of the specified sets of arguments. Further we successively describe our algorithms for approximation (HT-cross) and optimization (HTOpt) in the HT-format in details. We combine those two algorithms into HTBB algorithm, with overall structure described in Algorithm 1.

---

**Algorithm 1** High-level structure of HTBB optimization and approximation algorithm

---

**Require:** Black-box function $f$
**Ensure:** Extrema argument $I_0$ of $f$ and/or cores of HT-decomposition that approximates $f$
 1: Initialize a tree structure, indices sets, and indices values
 2: Start from the root node and set it as current
 3: **while** budget is not exhausted **do**
 4:    Choose an edge connected to the current node by the rule from Section 4.3
 5:    Apply index value update Algorithm 2 to update the indices values associated with the edge and current direction, as shown in Fig. 2 based on function $f$ values calculated on the corresponding indices
 6:    If needed, update the extrema argument $I_0$ if the extrema value was meet on the previous step
 7:    Go in the direction prescribed by the edge, set the new node as current
 8: **end while**
 9: Go to the cores building procedure described in Section 4.4 if needed

---

# 3 HIERARCHICAL TUCKER DECOMPOSITION

By Hierarchical Tucker (HT), we mean a tensor tree that is not necessarily balanced Ballani et al. (2013). Let us describe this concept in detail in the context of our work. HT is such a low-parameter decomposition of a $d$-way tensor, which is a hierarchical contraction of 3-way tensors and 2-way tensors, ordered in the form of a binary tree. Consider a binary tree — a graph without cycles, where every *node* (except the root one) has a *parent* and at most two *children*. In what follows, we consider trees where each node has either 2 or 0 children. We call a node without children a *leaf*. We denote the depth of the tree by $L$, and the number of nodes at level $l$ (starting from the root node) by $\lambda_l$; note that for a balanced tree, $\lambda_l = 2^{l-1}$ is satisfied. With each node, we associate a *core* tensor, i.e., a 2-way tensor with the leaves, and 3-way tensors with all others (for the root core we add a dummy dimension of the length 1). The number of leaves $d$ determines the dimensionality of the considered tensor $\mathcal{Y}$, which is represented in the described tree structure, i.e., $\mathcal{Y} \in \mathbb{R}^{N_1 \times N_2 \times \cdots \times N_d}$, where $N_j$ is the size of $j$th mode. The dimensions of the cores are as follows. Leaves dimensions correspond to the dimensionality of the tensor $\mathcal{Y}$: each core $\mathcal{G}_j^{(L)}$ that is associated with a leaf node with number $j$

---

[3]We consider here the tensor $\mathcal{H}$ as a function taking a discrete index as input and returning the real value

satisfies $\mathcal{G}_j^{(L)} \in \mathbb{R}^{r_j^{(L)} \times N_j}$. The dimensions of the non-leaves cores match the dimensions of their children: if core $\mathcal{G}_j^{(l)} \in \mathbb{R}^{r_{j_1}^{(l+1)} \times r_j^{(l)} \times r_{j_2}^{(l+1)}}$ for $1 \leq l < L$, then its child $\mathcal{G}_{j_1}^{(l+1)}$ and $\mathcal{G}_{j_2}^{(l+1)}$ have such dimensions that $\mathcal{G}_{j_1}^{(l+1)} \in \mathbb{R}^{r_1 \times r_{j_1}^{(l+1)} \times r_2}$ and $\mathcal{G}_{j_2}^{(l+1)} \in \mathbb{R}^{r_3 \times r_{j_2}^{(l+1)} \times r_4}$. The numbers $r_j^{(i)}$ are called *ranks* of the HT decomposition. The root core $\mathcal{G}_1^{(1)}$ has the following dimension: $\mathcal{G}_1^{(1)} \in \mathbb{R}^{r_{j_1}^{(2)} \times 1 \times r_{j_2}^{(2)}}$. In the case of notations related to tree nodes, the index at the top in parentheses denotes the level of the tree $l$, it varies from $1$ to $L$ ($L = \ln d$ for the balanced tree), and with the index at the bottom we denote the numbering within this level of the tree, this numbering is not fixed and is arbitrary.

To calculate the value of the tensor $\mathcal{Y}$ in the HT-format at an index $I$, we perform the following iterative procedure. We associate a vector $b_j^{(l)}$ with each node, which is recursively defined as

$$ b_j^{(l)} = \sum_{i=1}^{r_1} \sum_{k=1}^{r_2} \mathcal{G}_j^{(l)}[i, :, k] \cdot b_{j_1}^{(l+1)}[i] \cdot b_{j_2}^{(l+1)}[k], $$

where the vectors $b_{j_1}^{(l+1)}$ and $b_{j_2}^{(l+1)}$ are vectors associated with children of the current node; $r_1 = r_{j_1}^{(l+1)}$ and $r_2 = r_{j_2}^{(l+1)}$ are the corresponding ranks. For a leaf node, its corresponding vector $b_j^{(L)}$ depends on the given index $I$ as $b_j^{(L)} = \mathcal{G}_j^{(L)}[:, I[j]]$. Finally, the resulting tensor value at index $I$ is equal to the value of the single element of the vector $b_1^{(1)}$ associated with the root node, i.e., $\mathcal{Y}[I] = b_1^{(1)}[1]$. Note that this procedure is easily parallelized naturally since vectors $b$ of the same level in different parts of the tree are calculated independently.

# 4 DETAILS ON PROPOSED APPROACH

## 4.1 UPPER AND DOWN INDICES

The key concept that is used for both the approximation and optimization algorithm is to associate index sets with each link between nodes. Each link between node $D_m^{(l-1)}$ and its child $D_j^{(l)}$ have *down* $i_{l,j}^{\text{down}}$ and *upper* $i_{l,j}^{\text{up}}$ indices and corresponding values $v_{l,j}^{\text{down}}$ and $v_{l,j}^{\text{up}}$ of this indices. Since each link is unambiguously defined by the child node $D_j^{(l)}$ it is part of, the index notations are similar to this children node notation and sometimes we refer to these indices as being associated with the child node rather than a relation.

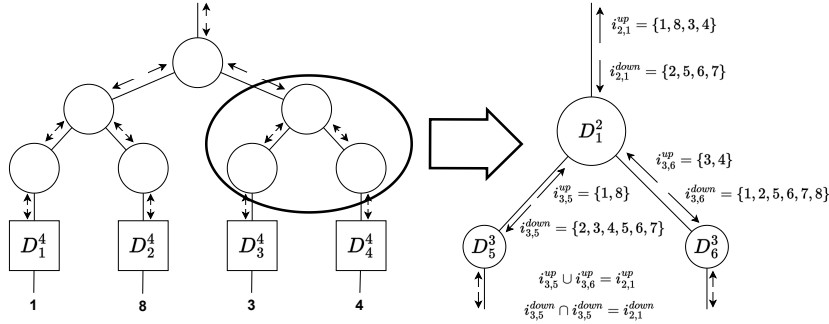

Figure 1: Examples of upper and down indices and their values for $\mathcal{Y} \in \mathbb{R}^{N_1 \times N_2 \times \dots N_8}$ with $N_1 = N_2 = N_5 = N_6 = N_7 = 2$, $N_3 = N_4 = 3$, and $N_8 = 10$.

Down $i_{l,j}^{\text{down}}$ and upper $i_{l,j}^{\text{up}}$ indices depend only on their position and are fixed during initial tree construction according to the following recursive rule. Each leaf node $D_j^{(L)}$ has upper index $i_{L,j}^{\text{up}} = \{j\}$ containing one element equal to the element number of the tensor index element that is associated with this leaf node. Each non-leaf node $D_j^{(l)}$ except the root one has an upper index consisting of the union of the elements of the upper indices of all its children: $i_{l,j}^{\text{up}} = i_{l+1,j_1}^{\text{up}} \cup i_{l+1,j_2}^{\text{up}}$. For all

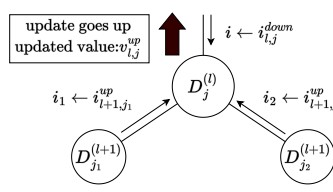 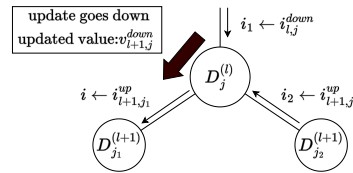

(a) Update upper indices values.      (b) Update down indices values.

Figure 2: Algorithm 2 inputs for the cases of upper and down indices values update. On the left: when updating upwards, the indices forming the rows of $A$ are calculated based on the upper indices on the links below ($i_1$ and $i_2$) and their values, and the indices $i$ (and their values $v$) forming the row of the matrix $A$ consists of the down indices of the link above and their values. The values of the upper indices associated with the link above are updated. On the right: similar updating but with slight changes occurs when moving downwards.

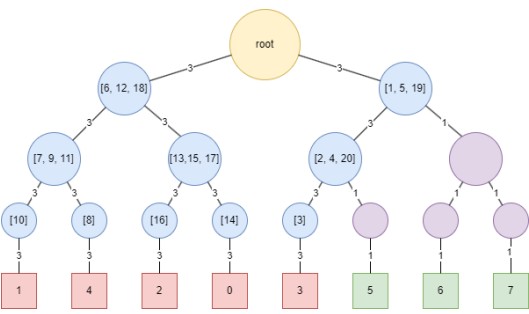

Figure 3: Examples of a path for the traversal procedure. The task is 5-dimensional, so indices 5, 6, and 7 (green boxes) as well as their parents (purple circles) are never visited. Ranks of all links, except for those leading to inactive indices, are equal to 3.

the cases, down indices are equal to the set difference between all tensor indices and upper indices: $i_{l,j}^{\text{down}} = \{1, 2, \ldots, d\} \setminus i_{l,j}^{\text{up}}$. Since the root node is not a child, we do not associate indexes with it. The down indices of the left child of the root node and their values are equal to the upper indices of the right child and their values, respectively, and vice versa. Please see Figure 1 for relevant illustration.

The values of the upper $v_{l,j}^{\text{up}}$ and down $v_{l,j}^{\text{down}}$ indices change dynamically and the manner and sequence of their change is the subject of this study. These values $v_{l,j}^{\text{up}}$ and $v_{l,j}^{\text{down}}$ represent sets of size equal to the rank, associated with the corresponding node: $\left| v_{l,j}^{\text{up}} \right| = \left| v_{l,j}^{\text{down}} \right| = r_j^{(l)}$. Each element of this set is a vector with values of indices stored in the corresponding ($i_{l,j}^{\text{up}}$ or $i_{l,j}^{\text{down}}$) index set. The main goal of the iterative search for index values (the detailed implementation of which will be described below) is to find the submatrix of maximum volume at the intersection of the given indices. Finding a submatrix of maximal volume serves two purposes: first, we can more accurately reconstruct the original matrix using it, and second, we expect that this matrix has elements close to maximal in modulo. Let us elaborate on the construction of this matrix.

Let $\mathcal{Y}^U(I)$ for the given index be the *unfolding* matrix of the $d$-way tensor $\mathcal{Y}$ in the given index $I = (k_1, k_2, \ldots, k_n)$, $1 \leq n \leq d$, if for all its elements holds

$$\mathcal{Y}^U(I)[\overline{i_{k_1} \cdots i_{k_n}}, \overline{i_{p_1} \cdots i_{p_{d-n}}}] = \mathcal{Y}[i_1, i_2, \ldots, i_d], \quad \{p_1, p_2, \ldots, p_{d-n}\} = \{1, 2, \ldots, d\} \setminus I.$$

By a line on a group of indices, we mean a multi-index composed of the given indices, *i. e.* the position of the corresponding sequence of indices in the list of all possible values. We do not fix a particular sorting type of this sequence (lexicographic order can be taken) since the rearrangement does not affect the rank of the matrix or the property of its submatrix of maximal volume.

For a non-leaf node $D_j^{(l)}$, the node up indices $i_{l,j}^{\text{up}}$ and up and down indices values $v_{l,j}^{\text{up}}$ and $v_{l,j}^{\text{down}}$ we can construct the unfolding $\mathcal{Y}_{l,j}^U$ as $Y_{l,j} = \mathcal{Y}^U(i_{l,j}^{\text{up}})$. If we consider submatrix $Y_{l,j}[v_{l,j}^{\text{up}}, v_{l,j}^{\text{down}}] \in$

---

**Algorithm 2** Indices values update algorithm.

---

**Require:** function $f$ for the $d$-way tensor value calculation; indices $i$, $i_1$ and $i_2$ such that $\text{set}(i \cup i_1 \cup i_2) = \{1, 2, \ldots, d\}$, and the corresponding indices values $v$, $v_1$ and $v_2$; maximum possible rank increment $\Delta r$; threshold for rank reduction $\epsilon$; transformation $T$.

**Ensure:** indices values $V$ of the index $I = i_1 \cup i_2$.

1: $r, r_1, r_2 = |v|, |v_1|, |v_2|$
2: *// First we build the tall matrix:*
3: $A \leftarrow zeros([r_1 \cdot r_2, r])$ *// Matrix with tensor values*
4: $J \leftarrow zeros(d)$ *// Integer index vector*
5: $F \leftarrow zeros([r_1 \cdot r_2, r_1 + r_2])$ *// Stores index candidates*
6: **for** $(j_1, j_2)$ in $\{1, 2, \ldots, r_1\} \times \{1, 2, \ldots, r_2\}$ **do**
7:     $J[i_1] \leftarrow v_1[j_1]$
8:     $J[i_2] \leftarrow v_2[j_2]$
9:     $F[\overline{j_1 j_2}, :] \leftarrow v_1[j_1] \cup v_2[j_2]$
10:    **for** $j$ in $\{1, 2, \ldots, r\}$ **do**
11:        $J[i] \leftarrow v[j]$
12:        $A[\overline{j_1 j_2}, j] \leftarrow f(J)$
13:    **end for**
14: **end for**
15: $A \leftarrow T(A)$ *// Apply point-wise transformation*
16: $\{Q, R, P\} \leftarrow \text{QRP}(A)$ *// Now use maxvol to select indices (QRP is QR with permutations)*
17: $r_\epsilon \leftarrow \max\{n \,|\, 1 \le n \le r, R[n, n]/R[0, 0] \ge \epsilon\}$
18: **if** $r_\epsilon < r$ **then**
19:    $\Delta r \leftarrow 0$ *// Decrease in rank occur, there is no point in raising it back again*
20:    $Q \leftarrow Q[:, 1{:}r_\epsilon]$
21: **end if**
22: $N \leftarrow \text{MaxVol}(Q, \Delta r)$ *// N is a vector of integers of length $r_o$, $r_\epsilon \le r_o \le r_\epsilon + \Delta r$* **return** $F[N, :]$ *// List of $r_o$ vectors of length $(r_1 + r_2)$*

---

$\mathbb{R}^{r \times r}$ of this matrix based on the values $v_{l,j}^{\text{up}}$ and $v_{l,j}^{\text{down}}$, where $r = r_j^{(l)}$ is the corresponding rank, when our goal is to make a volume of the matrix $Y_{l,j}[v_{l,j}^{\text{up}}, v_{l,j}^{\text{down}}]$ as large as possible by choosing indices values $v_{l,j}^{\text{up}}$ and $v_{l,j}^{\text{down}}$. Recall that the *volume* of any (tall) matrix $A$ is defined as

$$\text{vol}\, A = \sqrt{\det A^T A}, \quad A \in \mathbb{R}^{n \times m}, \quad n \ge m.$$

### 4.2 Index values update algorithm

While our method is running, we update all indices (both up and down), using the same procedure, as presented in Algorithm 2. Here the function $zeros$ reserves the specified number of elements for a vector, matrix, etc. The QRP function returns a QR decomposition with permutations (*i. e.*, the elements on the diagonal of $R$ do not decrease; we use the implementation from the Python package `scipy`). The operation of the algorithm can be briefly described as follows. We construct a tall matrix, whose rows correspond to the tensor product of index values, which are conditionally called "incoming" and columns to "outgoing" ones. Then, using the MaxVol procedure, we select rows from this matrix so that the submatrix corresponding to them is of quasi-maximal volume, and the index values corresponding to these rows are returned.

**Algorithm input indices.** The "incoming" $i_1$, $i_2$, and "outgoing" $i$ input indices in our algorithm depend on the indices that are updated at each step (see Figure 2). Namely, if we update the upper indices values $v_{l,j}^{\text{up}}$ for some node $D_j^{(l)}$, then "incoming" indices are the upper indices of children $D_{j_1}^{(l+1)}$ and $D_{j_2}^{(l+1)}$ of this node: $i_1 \leftarrow i_{l+1,j_1}^{\text{up}}$, $i_2 \leftarrow i_{l+1,j_2}^{\text{up}}$. The "output" indices are down indices of the node $D_j^{(l)}$: $i \leftarrow i_{l,j}^{\text{down}}$ (see Figure 2a). If, in turn, we update the values of down indices for the link that connects parent $D_j^{(l)}$ and child $D_{j_1}^{(l+1)}$, then for "incoming" indices we have: $i_1 \leftarrow i_{l,j}^{\text{down}}$, $i_2 \leftarrow i_{l+1,j_2}^{\text{up}}$, where $j_2$ is the number of another child $D_{j_2}^{(l+1)}$ of the node $D_j^{(l)}$ (which

differs from the original child $D_{j_1}^{(l+1)}$). The upper indices of the given link are the "output" ones: $i \leftarrow i_{l+1,j_1}^{\mathrm{up}}$ (see Figure 2b).

**Transformation of the tensor values.** The point-wise transformation is needed when we search for the minimum. In this case, we can transform tensor values by any monotonic decreasing function $T$. In our experiments, we use the following adaptive (*i. e.* its parameters dependent on the given data batch) transformation $T(x) = \exp(-(x-x_0)/\sigma)$, $x_0 = \mathrm{mean}(x)$, $\sigma = \mathrm{std}(x)$, where $\mathrm{mean}(x)$ and $\mathrm{std}(x)$ are the sample mean and sample variance of the set of numbers respectively. When searching for the maximum value, we do a similar transformation (note that transformation can be avoided in this case; however, we apply it for greater stability of the method) $T(x) = \exp((x - x_0)/\sigma)$.

**MaxVol procedure.** The $\mathrm{MaxVol}$ procedure in our algorithm is the so-called *rectangular* maximum volume search method Mikhalev & Oseledets (2018). Note that it can return not only square matrices but also rectangular matrices, making a decision on the number of returned rows based on a heuristic procedure based on the possible increase in volume when adding a candidate row and the given tuning parameters. In our numerical experiments, we allowed to expand output index set by at most $\Delta r = 1$ element, so the ranks grew by at most 1 per pass.

### 4.3 TRAVERSAL PROCEDURE

When updating indices, we walk sequentially to the neighboring (linked) node, going back only if we reach a leaf node. At each visited node, we increment its visit counter by one, whereas at the beginning all counters were reset to zero. When we pass through an edge, we update only one set of index values at a time: if we go from parent to child, we update down indices values; if we go from child to parent, we update upper indices values.

To decide which of the two nodes to take the next step to (in case there are two options), we count the average number of visits in each part of the tree that separates each of the two paths. Namely, we cut the edge that was traveled last, and we cut the edges connecting the current node to the two candidate nodes. Since by definition there are no loops in the tree and each edge is a cut edge, we get three components of connectedness. Then we calculate the average number of visits (the sum of the number of visits on all nodes divided by the number of nodes) in each of the two connectivity components and go where the number is smaller. If the average number of visits is close, namely, they differ by no more than a given $\alpha$ value, then we randomly choose one side to go to. Please, see Figure 3 for an example of path, where each number in a list inside a node (blue circle) represents the number of steps when updates occur in this node.

### 4.4 CORES BUILDING

After all indices are found by the search procedure described above, we can build all cores based on these indices. First, consider a leaf node $D_j^{(L)}$, and let its down indices are $i_{L,j}^{\mathrm{down}}$ and the values of these indices are $v_{L,j}^{\mathrm{down}}$ (recall, that $\{j\} \cup i_{L,j}^{\mathrm{down}} = \{1, 2, \ldots, d\}$). Then the core, associated with this node is calculated as follows. First, we form a matrix $V$ of values of the BB using these indices

$$V[i, k] = f(I_{ik}) \quad \text{with} \quad I_{ik}[j] = i, \quad I_{ik}[i_{L,j}^{\mathrm{down}}] = v_{L,j}^{\mathrm{down}}[k], \quad \forall 1 \leq i \leq N_j, \ 1 \leq k \leq r_j^{(L)},$$

and then we let the core $\mathcal{G}_j^{(L)}$ be the transposed factor $Q$ of the QR-decomposition of this matrix $V$

$$\mathcal{G}_j^{(L)} = Q^{\mathbf{T}}, \quad \text{where} \quad \{Q, R\} = \mathrm{QR}(V).$$

For the non-leaf and non-root node $D_j^{(l)}$ we perform a similar procedure. Let $i_{l,j}^{\mathrm{down}}$ and $v_{l,j}^{\mathrm{down}}$ be its down indices and its down indices value, respectively. Let $i_{l+1,j_c}^{\mathrm{up}}$ and $v_{l+1,j_c}^{\mathrm{up}}$ be upper indices and upper indices value, respectively, for the $c$th child of this node, where $c = 1, 2$ (recall, that $i_{l+1,j_1}^{\mathrm{up}} \cup i_{l+1,j_2}^{\mathrm{up}} \cup i_{l,j}^{\mathrm{down}} = \{1, 2, \ldots, d\}$). Then we first build the matrix $V$

$$V[\overline{in}, k] = f(I_{ink}) \quad \text{with} \quad I_{ink}[i_{l+1,j_1}^{\mathrm{up}}] = v_{l+1,j_1}^{\mathrm{up}}[i], \quad I_{ink}[i_{l+1,j_2}^{\mathrm{up}}] = v_{l+1,j_2}^{\mathrm{up}}[n],$$
$$I_{ink}[i_{l,j}^{\mathrm{down}}] = v_{l,j}^{\mathrm{down}}[k], \quad \forall 1 \leq i \leq r_{j_1}^{(l+1)}, \ 1 \leq n \leq r_{j_2}^{(l+1)}, \ 1 \leq k \leq r_j^{(l)},$$

and then we let the values of the core $\mathcal{G}_j^{(l)}$ be the "reshaped" values of the factor $Q$ of the QR-decomposition of $V$

$$\mathcal{G}_j^{(l)}[i,\,k,\,n] = Q[\overline{in},\,k], \text{ where } \{Q,\,R\} = \mathrm{QR}(V).$$

Finally, for the root node $D_1^{(1)}$, we let the values of the assigned core be the values of the given BB in the corresponding points. Namely, let $i_{2,j_c}^{\mathrm{up}}$ and $v_{2,j_c}^{\mathrm{up}}$ be upper indices and upper indices value for the $c$th child of the root node, $c = 1, 2$. Then for all $1 \le i \le r_{j_1}^{(2)}$, $1 \le n \le r_{j_2}^{(2)}$ we have

$$\mathcal{G}_1^{(1)}[i,\,1,\,n] = Q[i,\,n], \quad Q[i,\,n] = f(I_{in}) \text{ with } I_{in}[i_{2,j_1}^{\mathrm{up}}] = v_{2,j_1}^{\mathrm{up}}[i], \ I_{in}[i_{2,j_2}^{\mathrm{up}}] = v_{2,j_2}^{\mathrm{up}}[n].$$

Note, that due to this procedure, the obtained cores are orthogonalized and, therefore, their maximum modulus values are moderate.

### 4.5 COMPLEXITY ESTIMATION

One can see, that Alg. 2 require the total number $N_1$ of function $f$ call equals to $N_1 = \text{size\_of}(A) = rr_1r_2$, where size\_of$(A)$ is the total number of elements in the matrix $A$ (we use the notation from the Alg). Roughly speaking, for one step we need $O(r_m^3)$ calls, where $r_m$ is the maximum rank. One complete sweep, where we move from one leaf node and return to the same node, requires at most $kd$ such updates, where constant $k$ depends only on the chosen traversal algorithm. For example, for the one we used in the experiments and described in Sec. 4.3, $k = 2$. Thus, in total, the number of requests to the black box is $O(nr_m^3d)$, where $n$ is the number of sweeps.

## 5 RELATED WORK

In many practical situations, the problem-specific target function is not differentiable, too complex, or its gradients are not helpful due to the non-convex nature of the problem, and it has to be treated as a black box (BB). In this case, two important problems naturally arise: approximation Bhosekar & Ierapetritou (2018) and optimization Alarie et al. (2021). The approximation carried out in the offline phase allows us to build a surrogate (simplified) model of the BB, which can then be used in the online phase to quickly calculate its values and various characteristics. In the multidimensional case, it becomes difficult to construct a surrogate model, and low-rank tensor approximations are often the most effective. Several recent works Kapushev et al. (2020); Ahmadi-Asl et al. (2021); Chertkov et al. (2023b) proposed various new algorithms based on the TT-decomposition for approximating high-dimensional functions. If we have access to the BB and can perform dynamic queries, then the powerful TT-cross method Oseledets & Tyrtyshnikov (2010) is often used, and if only a training dataset is available, then the TT-ALS method Holtz et al. (2012) is preferred. In this work, we consider the case of adaptive queries to the BB, so we select the TT-cross method as the main baseline for the approximation problem.

Gradients are not available for the BB, so only gradient-free methods can be used for the optimization problem. Particle Swarm Optimization (PSO) Kennedy & Eberhart (1995) and Simultaneous Perturbation Stochastic Approximation (SPSA) Maryak & Chin (2001) are rather useful methods in this case. There is also a large variety of other heuristic methods for finding the global extremum. Recently, the TT-decomposition has been actively used for black-box optimization, since it turns out to be more effective than standard approaches in the multidimensional case. An iterative method TTOpt based on the maximum volume approach is proposed in the work Sozykin et al. (2022). The authors applied this approach to the problem of optimizing the weights of neural networks in the framework of reinforcement learning problems in Sozykin et al. (2022) and to the QUBO problem in Nikitin et al. (2022). A similar optimization approach was also considered in Selvanayagam et al. (2022) and Shetty et al. (2016). One more promising algorithm, Optima-TT, which is based on the probabilistic sampling from the TT-tensor, was proposed in recent work Chertkov et al. (2023a). We also note the work Soley et al. (2021), where an optimization method based on the iterative power algorithm in terms of the quantized version of the TT-decomposition is proposed. As a result, we consider classical PSO and SPSA methods as well as the TTOpt method as baselines for the optimization problem.

Table 1: Approximation relative error for the HTBB and TT-cross applied to all considered $d = 256$-dimensional benchmarks. The reported values are averaged over 10 independent runs.

| BENCHMARK | HTBB | TT-CROSS |
|---|---|---|
| ALPINE | **2.83E-15** | 1.73E-02 |
| CHUNG | **7.87E-03** | 2.86E-02 |
| DIXON | **5.65E-03** | 1.00E-01 |
| GRIEWANK | **2.83E-15** | 1.43E-02 |
| PATHOLOGICAL | **3.92E-02** | 1.08E-01 |
| PINTER | **1.23E-02** | 1.47E-02 |
| QING | **3.67E-02** | 4.87E-02 |
| RASTRIGIN | **1.01E-14** | 1.47E-02 |
| SCHAFFER | **1.87E-02** | 1.88E-02 |
| SCHWEFEL | **3.39E-14** | 6.31E-01 |
| SPHERE | **1.20E-14** | 1.44E-02 |
| SQUARES | **1.07E-14** | 1.77E-02 |
| TRIGONOMETRIC | **2.76E-02** | 4.82E-02 |
| WAVY | **8.56E-05** | 2.46E-03 |

Table 2: Approximation relative error for the HTBB applied to all considered 512 and 1024-dimensional benchmarks. The reported values are averaged over 5 independent runs.

| BENCHMARK | $d = 512$ | $d = 1024$ |
|---|---|---|
| ALPINE | 4.92E-15 | 3.81E-04 |
| CHUNG | 7.86E-03 | 7.64E-03 |
| DIXON | 3.75E-03 | 2.83E-03 |
| GRIEWANK | 1.37E-14 | 3.16E-14 |
| PATHOLOGICAL | 3.80E-02 | 3.76E-02 |
| PINTER | 8.80E-03 | 8.38E-03 |
| QING | 1.85E-02 | 1.60E-02 |
| RASTRIGIN | 1.63E-14 | 1.02E-04 |
| SCHAFFER | 1.94E-02 | 1.52E-02 |
| SCHWEFEL | 2.59E-13 | 1.23E-13 |
| SPHERE | 1.16E-14 | 4.58E-14 |
| SQUARES | 1.08E-14 | 2.38E-14 |
| TRIGONOMETRIC | 2.74E-02 | 2.38E-02 |
| WAVY | 1.18E-04 | 3.38E-04 |

Table 3: Minimization results for the HTBB, TTOpt, One+One, SPSA, and PSO applied to 256-dimensional benchmarks. The reported values are averaged over 10 independent runs.

| BENCHMARK | HTBB | TTOPT | ONE+ONE | SPSA | PSO |
|---|---|---|---|---|---|
| ALPINE | **6.75E+01** | 4.48E+02 | 3.66E+02 | 3.99E+02 | 4.76E+02 |
| CHUNG | **1.45E+06** | 7.74E+07 | 1.48E+06 | 1.54E+06 | 6.98E+07 |
| DIXON | **1.89E+06** | 1.99E+08 | 2.33E+06 | 3.20E+06 | 2.68E+08 |
| GRIEWANK | **3.11E+01** | 2.21E+02 | 3.19E+01 | 3.12E+01 | 2.09E+02 |
| PATHOLOGICAL | **6.97E+01** | 1.02E+02 | 1.14E+02 | 9.32E+01 | 1.06E+02 |
| PINTER | **5.17E+05** | 1.19E+06 | 5.67E+05 | 5.94E+05 | 1.51E+06 |
| QING | **4.99E+06** | 2.98E+12 | 8.47E+10 | 1.26E+12 | 1.76E+12 |
| RASTRIGIN | **9.19E+02** | 3.61E+03 | 1.09E+03 | 9.31E+02 | 3.72E+03 |
| SCHAFFER | **9.87E+01** | 1.15E+02 | 1.06E+02 | 1.02E+02 | 1.20E+02 |
| SCHWEFEL | **-3.85E+02** | -1.77E+02 | -1.92E+02 | -1.89E+02 | -1.38E+02 |
| SPHERE | **3.16E+02** | 2.30E+03 | 3.24E+02 | 3.17E+02 | 2.19E+03 |
| SQUARES | **1.55E+05** | 7.37E+05 | 1.57E+05 | **1.55E+05** | 1.02E+06 |
| TRIGONOMETRIC | **8.72E+04** | 9.30E+06 | 2.62E+05 | 1.77E+07 | 1.01E+07 |
| WAVY | **3.19E-01** | 6.21E-01 | 3.64E-01 | 3.22E-01 | 6.36E-01 |

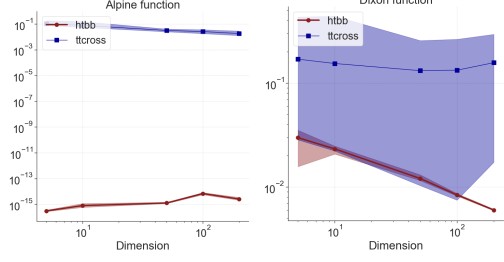

(a) Approximation results for Alpine and Dixon functions for cases of dimensions 5, 10, 50, 100, and 200. For both methods, we plot the relative error of the solution averaged over 10 runs with a solid line and fill in the area between the worst and best result with the same color.

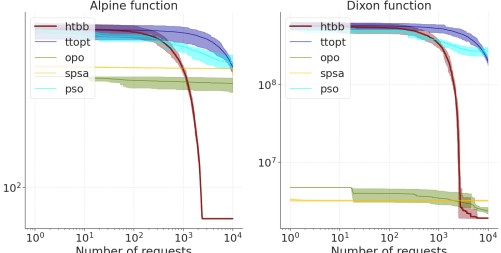

(b) Minimization results for Alpine and Dixon functions. For each of the optimizers, we plot the value of the solution averaged over 10 runs with a solid line and fill in the area between the worst and best result with the same color.

Figure 4: Approximation and optimization results.

## 6 NUMERICAL EXPERIMENTS

To demonstrate the effectiveness of the proposed HTBB approach, we select 14 popular 256-dimensional benchmarks Jamil & Yang (2013); Vanaret et al. (2020); Dieterich & Hartke (2012), which correspond to analytical functions with complex landscape and are described in detail in Table 4 from Appendix. For each benchmark, we fix the input dimension at 256 and consider the approximation and optimization problem in the black-box settings for the tensor that arises when the corresponding function is discretized on a Chebyshev grid with 8 nodes in each dimension. In all cases, we limited the budget (the number of requests to the BB) to $10^4$, and the HT-rank was 2.

In Appendix, we also present the results of additional numerical experiments for multidimensional black-box approximation (we consider various neural network architectures as additional baselines) and optimization (we consider four challenging problems of maximization of the cumulative reward for reinforcement learning agents by the proposed method HTBB and various baselines).

### 6.1 MULTIDIMENSIONAL APPROXIMATION

For each 256-dimensional benchmark we perform the approximation with the proposed HTBB method and compare it with the TT-cross method,[4] constrained by the same budget ($10^4$ requests to BB). The relative $L2$ errors on test sets of $10^4$ random points which were generated for each benchmark are reported in Table 1 (the computations were repeated 10 times for both methods and the averaged results are presented).

Also in Figure 4a we provide a graphical comparison of the results for two benchmarks for the case of different values of the problem dimension (5, 10, 50, 100, 200). As follows from the presented results, for all problems our method turns out to be more accurate than the baseline, and in some cases its accuracy turns out to be many orders of magnitude higher. For the case of higher dimensions for the considered problem classes, running the TT-cross method leads to failures in software implementation due to instability, while our approach remains stable and gives high accuracy, as follows from values reported in Table 2 for dimensions 512 and 1024.

### 6.2 MULTIDIMENSIONAL OPTIMIZATION

For each 256-dimensional benchmark we perform the optimization (namely the search for a global minimum) with the proposed HTBB method. We consider as baselines the tensor-based optimization method TTOpt[5] and three popular gradient-free optimization algorithms from the nevergrad framework Bennet et al. (2021):[6] One+One, SPSA, and PSO. The limit on the number of requests to the objective function was fixed at the value $10^4$. The calculations were repeated 10 times for all methods and the averaged results are presented in Table 3. Also in Figure 4b we show the convergence plots for two benchmarks. As follows from the reported values, HTBB, in contrast to alternative approaches, gives a consistently top result for all model problems.

## 7 CONCLUSIONS

In this work, we presented a new method HTBB for simultaneously solving the problem of multidimensional approximation and gradient-free optimization for functions given in the form of a black box. Our approach is based on the low-rank hierarchical Tucker decomposition, which makes it especially effective in the multidimensional case. The key features of the presented work are a) using the MaxVol algorithm which allows efficiently finding the required indices and b) using the sequential traversal of cores, allowing to move to one of the neighboring nodes and making it more efficient to find indexes that need updating.

The HTBB method can be applied to a wide class of practically significant problems, including optimal control and various machine learning applications. As future work, we point out the possibility of a rather simple extension on the HT-structure of the algorithms that now exist for the TT-decomposition: rounding, orthogonalization, search for the maximum element by the top-k-like methods, etc.

---

[4]We used the TT-cross method from https://github.com/AndreiChertkov/teneva.

[5]We used the implementation of the TTOpt https://github.com/AndreiChertkov/ttopt.

[6]See https://github.com/facebookresearch/nevergrad.

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

## A  DESCRIPTION OF THE USED BENCHMARKS

In all numerical experiments on approximation and optimization, we used a single set of benchmarks Jamil & Yang (2013); Vanaret et al. (2020); Dieterich & Hartke (2012) representing discretized analytical functions of a multidimensional argument. A description of the functions is presented in Table 4. We note that they have a complex landscape and are often used when testing surrogate modeling and optimization algorithms.

## B  MULTIDIMENSIONAL APPROXIMATION WITH NEURAL NETWORKS

In the main text, we considered the tensor-based TT-cross method as a baseline in the case of approximation problems. To demonstrate that the proposed method HTBB outperforms other modern approaches, in this section we present approximation results obtained using the well-known CatBoost model Prokhorenkova et al. (2018) and neural networks with relu-type nonlinearities and fully connected 4 layers ("MLP-1" in the results presented below), 3 layers ("MLP-2"), and 5 layers ("MLP-3"), while in all cases the number of neurons in the inner layer is about $1000$.

For the same set of $14$ analytical functions and a generated training dataset of size 10k, we present the results for dimensions $256$, $512$, and $1024$ in Table 5, Table 6, and Table 7 respectively (for ease

Table 4: Benchmark functions for performance analysis of the proposed method.

| FUNCTION | BOUNDS | ANALYTICAL FORMULA |
|---|---|---|
| ALPINE | $[-10, 10]$ | $\sum_{i=1}^{d} \lvert x_i \sin x_i + 0.1 x_i \rvert$ |
| CHUNG | $[-10, 10]$ | $\left( \sum_{i=1}^{d} x_i^2 \right)^2$ |
| DIXON | $[-10, 10]$ | $F(\boldsymbol{x}) = (x_1 - 1)^2 + \sum_{i=2}^{d} i \cdot \left( 2x_i^2 - x_{i-1} \right)^2$ |
| GRIEWANK | $[-100, 100]$ | $\sum_{i=1}^{d} \frac{x_i^2}{4000} - \prod_{i=1}^{d} \cos\left( \frac{x_i}{\sqrt{i}} \right) + 1$ |
| PATHOLOGICAL | $[-100, 100]$ | $\sum_{i=1}^{d-1} \left( 0.5 + \frac{\sin^2 \sqrt{100 x_i^2 + x_{i+1}^2} - 0.5}{1 + 0.001 \left( x_i^2 - 2x_i x_{i+1} + x_{i+1}^2 \right)^2} \right)$ |
| PINTER | $[-10, 10]$ | $\sum_{i=1}^{d} \left( i x_i^2 + 20 i \sin^2 A_i + i \log_{10}\left( 1 + i B_i^2 \right) \right)$, WHERE $A_i = x_{i-1} \sin x_i + \sin x_{i+1}$, $B_i = x_{i-1}^2 - 2x_i + 3x_{i+1} - \cos x_i + 1$) WITH $x_0 = x_d$ AND $x_{d+1} = x_1$ |
| QING | $[0, 500]$ | $F(\boldsymbol{x}) = \sum_{i=1}^{d} \left( x_i^2 - i \right)^2$ |
| RASTRIGIN | $[-5.12, 5.12]$ | $A \cdot d + \sum_{i=1}^{d} \left( x_i^2 - A \cdot \cos\left( 2\pi x_i \right) \right)$, WHERE $A = 10$ |
| SCHAFFER | $[-100, 100]$ | $\sum_{i=1}^{d-1} \left( 0.5 + \frac{\sin^2 \left( \sqrt{x_i^2 + x_{i+1}^2} \right) - 0.5}{\left( 1 + 0.001 \cdot (x_i^2 + x_{i+1}^2) \right)^2} \right)$ |
| SCHWEFEL | $[0, 500]$ | $-\frac{1}{d} \sum_{i=1}^{d} x_i \cdot \sin\left( \sqrt{\lvert x_i \rvert} \right)$ |
| SPHERE | $[-5.12, 5.12]$ | $\sum_{i=1}^{d} x_i^2$ |
| SQUARES | $[-10, 10]$ | $\sum_{i=1}^{d} i x_i^2$ |
| TRIGONOMETRIC | $[0, \pi]$ | $\sum_{i=1}^{d} \left( d - \sum_{j=1}^{d} \cos x_j + i(1 - \cos x_i - \sin x_i) \right)^2$ |
| WAVY | $[-\pi, \pi]$ | $1 - \frac{1}{d} \sum_{i=1}^{d} \cos\left( k x_i \right) \cdot e^{-\frac{x_i^2}{2}}$, WHERE $k = 10$ |

Table 5: Relative error of the result in additional experiments on approximation of multidimensional functions for dimension 256.

| Name | MLP-1 | MLP-2 | MLP-3 | CatBoost | HTBB |
|---|---|---|---|---|---|
| Alpine | 3.56E-02 | 3.57E-02 | 3.67E-02 | 3.78E-02 | **2.83E-15** |
| Chung | 8.07E-02 | 2.26E-01 | 8.14E-02 | 6.80E-02 | **7.87E-03** |
| Dixon | 7.37E-02 | 8.22E-01 | 7.14E-02 | 4.15E-02 | **5.65E-03** |
| Griewank | 9.76E-03 | 9.78E-03 | 9.21E-03 | 1.76E-02 | **2.83E-15** |
| Pathological | 5.77E-02 | 5.97E-02 | 5.86E-02 | **3.21E-02** | 3.92E-02 |
| Pinter | 2.06E-02 | 2.60E-02 | 2.07E-02 | 2.54E-02 | **1.23E-02** |
| Qing | 5.35E-02 | 5.38E-02 | 5.35E-02 | **6.13E-03** | 3.67E-02 |
| Rastrigin | 2.36E-02 | 2.40E-02 | 2.34E-02 | 3.41E-02 | **1.01E-14** |
| Schaffer | 5.79E-02 | 5.83E-02 | 6.55E-02 | 4.38E-02 | **1.87E-02** |
| Schwefel | 2.02E-02 | 2.27E-02 | 2.28E-02 | 1.81E-02 | **3.39E-14** |
| Sphere | 2.45E-02 | 2.34E-02 | 2.32E-02 | 3.33E-02 | **1.20E-14** |
| Squares | 2.80E-02 | 3.78E-02 | 2.60E-02 | 3.08E-02 | **1.07E-14** |
| Trigonometric | 1.17E-01 | 5.90E-01 | 1.18E-01 | 7.98E-02 | **2.76E-02** |
| Wavy | 3.85E-02 | 4.39E-02 | 3.95E-02 | 1.51E-02 | **8.56E-05** |

Table 6: Relative error of the result in additional experiments on approximation of multidimensional functions for dimension $512$.

| Name | MLP-1 | MLP-2 | MLP-3 | CatBoost | HTBB |
|---|---|---|---|---|---|
| Alpine | 2.60E-02 | 2.60E-02 | 2.68E-02 | 3.03E-02 | **4.92E-15** |
| Chung | 5.75E-02 | 7.11E-01 | 5.74E-02 | 6.26E-02 | **7.86E-03** |
| Dixon | 5.26E-02 | 9.77E-01 | 5.24E-02 | 4.17E-02 | **3.75E-03** |
| Griewank | 9.08E-03 | 1.40E-02 | 1.03E-02 | 2.17E-02 | **1.37E-14** |
| Pathological | 4.70E-02 | 4.62E-02 | 4.95E-02 | **2.47E-02** | 3.80E-02 |
| Pinter | 1.71E-02 | 1.89E-02 | 1.48E-02 | 2.18E-02 | **8.80E-03** |
| Qing | 3.34E-02 | 3.94E-01 | 3.36E-02 | **3.21E-03** | 1.85E-02 |
| Rastrigin | 1.72E-02 | 1.79E-02 | 1.72E-02 | 3.16E-02 | **1.63E-14** |
| Schaffer | 4.29E-02 | 4.27E-02 | 4.34E-02 | 3.15E-02 | **1.94E-02** |
| Schwefel | 1.94E-02 | 1.75E-02 | 1.47E-02 | 1.78E-02 | **2.59E-13** |
| Sphere | 1.69E-02 | 1.73E-02 | 1.76E-02 | 3.19E-02 | **1.16E-14** |
| Squares | 2.35E-02 | 2.82E-02 | 2.18E-02 | 3.03E-02 | **1.08E-14** |
| Trigonometric | 8.24E-02 | 9.68E-01 | 8.20E-02 | 6.83E-02 | **2.74E-02** |
| Wavy | 2.84E-02 | 3.24E-02 | 2.92E-02 | 1.39E-02 | **1.18E-04** |

of comparison, we also duplicate the results for our method, which were already presented in the main text). As follows from the reported results, for almost all benchmarks our method HTBB shows significantly better result than the neural network-based approach.

## C  MULTIDIMENSIONAL OPTIMIZATION OF THE CUMULATIVE REWARD FOR REINFORCEMENT LEARNING AGENTS

We conduct a series of additional numerical experiments for a more explicit demonstration of the possibility of effective application of the proposed method HTBB to practical problems. As in the work Sozykin et al. (2022), we consider four challenging problems of maximization of the cumulative reward for reinforcement learning (RL) agents: InvertedPendulum, Swimmer, Lunar Lander, and Half Cheetah from the well-known Mujoco / OpenAI-GYM collection Brockman (2016). The policy is represented by a neural network with three layers and tanh activations, and its parameters are discretized on a grid with limits from $-1$ to $+1$ and 3 nodes. Thus, as in the work Sozykin et al. (2022), we obtain a discrete on-policy learning problem (search for the values of the neural network parameters that lead to the maximum reward).

Table 7: Relative error of the result in additional experiments on approximation of multidimensional functions for dimension 1024.

| Name | MLP-1 | MLP-2 | MLP-3 | CatBoost | HTBB |
|---|---|---|---|---|---|
| Alpine | 2.03E-02 | 1.81E-02 | 1.84E-02 | 2.34E-02 | **3.81E-04** |
| Chung | 3.99E-02 | 9.37E-01 | 4.01E-02 | 5.12E-02 | **7.64E-03** |
| Dixon | 1.00E+00 | 1.00E+00 | 1.00E+00 | 3.74E-02 | **2.83E-03** |
| Griewank | 1.13E-02 | 8.62E-03 | 9.64E-03 | 2.09E-02 | **3.16E-14** |
| Pathological | 3.35E-02 | 3.61E-02 | 3.27E-02 | **1.80E-02** | 3.76E-02 |
| Pinter | 1.24E-02 | 1.37E-02 | 1.18E-02 | 1.85E-02 | **8.38E-03** |
| Qing | 2.19E-02 | 9.58E-01 | 2.20E-02 | **1.41E-03** | 1.60E-02 |
| Rastrigin | 1.22E-02 | 1.20E-02 | 1.27E-02 | 2.56E-02 | **1.02E-04** |
| Schaffer | 3.51E-02 | 3.16E-02 | 3.15E-02 | 2.22E-02 | **1.52E-02** |
| Schwefel | 1.73E-02 | 1.26E-02 | 1.05E-02 | 1.45E-02 | **1.23E-13** |
| Sphere | 1.22E-02 | 1.22E-02 | 1.61E-02 | 2.57E-02 | **4.58E-14** |
| Squares | 2.02E-02 | 2.09E-02 | 1.69E-02 | 2.67E-02 | **2.38E-14** |
| Trigonometric | 1.00E+00 | 1.00E+00 | 1.00E+00 | 5.18E-02 | **2.38E-02** |
| Wavy | 2.36E-02 | 2.23E-02 | 1.96E-02 | 1.16E-02 | **3.38E-04** |

Table 8: Maximization results (reward values) for the HTBB, TTOpt, SPSA, and PSO applied to all considered RL benchmarks.

| BENCHMARK | HTBB | TTOPT | SPSA | PSO |
|---|---|---|---|---|
| INV. PENDULUM | **1.0E+03** | **1.0E+03** | 3.2E+01 | **1.0E+03** |
| SWIMMER | **3.5E+02** | **3.5E+02** | 2.9E+01 | 3.1E+02 |
| LUNAR LANDER | **2.6E+02** | 1.4E+01 | -2.9E+02 | 6.8E+01 |
| HALF CHEETAH | **1.9E+03** | **1.9E+03** | -1.5E+00 | 1.7E+03 |

Table 9: Computation times (in seconds) for the HTBB, TTOpt, SPSA, and PSO applied to all considered RL benchmarks.

| BENCHMARK | HTBB | TTOPT | SPSA | PSO |
|---|---|---|---|---|
| INV. PENDULUM | **2.7E+02** | 8.7E+02 | 5.2E+02 | 7.3E+03 |
| SWIMMER | 1.5E+04 | 1.5E+04 | 1.5E+04 | **1.4E+04** |
| LUNAR LANDER | **3.6E+03** | 9.0E+03 | 7.5E+03 | 4.7E+03 |
| HALF CHEETAH | **1.2E+04** | **1.2E+04** | 1.4E+04 | **1.2E+04** |

The computation results for the budget value equal 100 K are presented in In Table 8 (the final reward at the end of training for each of the optimization methods and each of the considered problems), and Table 9 (the related computation times). Figure 5 shows the corresponding convergence graphs for each of the methods. As follows from the reported values, for all four considered problems our method HTBB leads to the best result and shows the best performance (in the context of computation time) compared to baselines.

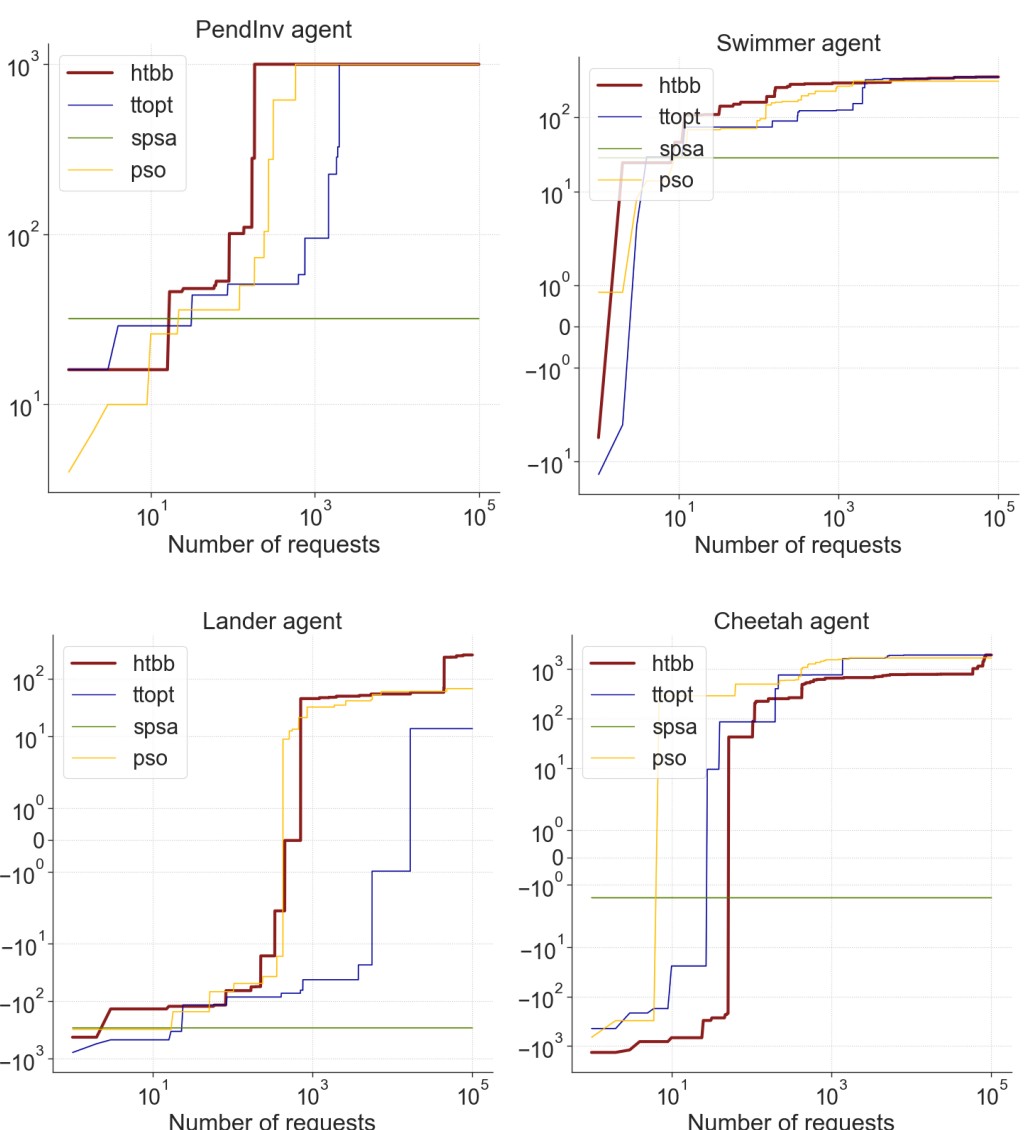

Figure 5: Maximization results (reward values) for the HTBB, TTOpt, SPSA, and PSO applied to all considered RL benchmarks.

