# OpenReview forum: "Black-Box Approximation and Optimization with Hierarchical Tucker Decomposition"
_ICLR.cc/2025/Conference — Submitted to ICLR 2025_

### Official Review · Reviewer_hayQ · 2024-10-29

**Soundness:** 3
**Presentation:** 2
**Contribution:** 2
**Rating:** 5
**Confidence:** 2

**Summary:**

This paper is concerned with the approximation and maximization (or minimization) of a real-valued function defined on a subset of $\mathbb{R}^d$. The function is considered as being a "black box", meaning its properties are unknown, but one can evaluate it over a predefined discretization grid, implicitly defining a tensor (of point function evaluations). Specifically, the authors propose extensions of the TT-Cross and TTOpt algorithms, which rely on the so-called tensor train (TT) tensor format, to a more general hierarchical Tucker (HT) format that is specified by a tree topology which indicates how the cores (factors) involved in the low-rank approximation are connected. The main technical difficulty seems to be how to navigate that tree for selecting a subset of indices used for building the approximation.

**Strengths:**

S1) The paper generalizes tensor-tain-based function approximation algorithms to more general hierarchical Tucker structures.

S2) Its simulation results convincingly illustrate the benefits of the proposed approach in terms of function approximation and maximization performance, presumably thanks to the higher degree of flexibility of HT in comparison with TT.

**Weaknesses:**

W1) Despite the proposed algorithms being potentially useful for practitioners, in my view the contribution of this paper is fairly incremental, as it extends some well-known algorithms to a different tensor format, but without introducing new ideas, insights or relevant theoretical results.

W2) The paper is fairly hard to read for non-experts, which I believe will be the case for the majority of the ICLR audience. In that regard:
- A brief description of the rationale behind the TT-Cross and TTOpt should be given, so that the readers understand what the authors are trying to implement. This will make the paper minimally self-contained, since the extension of these algorithms is the main goal here.
- Related do that point, is not quite clear what is the output of Algorithm 1. It seems to be the cores that will define the approximation, but at the same time this algorithm is expected to yield the max (or min) of the function, as announced in the paper.

W3) The paper should also clearly state exactly where the novelty is in terms of the introduced algorithms.

W4) A careful revision is needed for improving English usage. Examples:
- "possible maximum rank increment" -> maximum possible rank increment
- "then we go to the random side" -> then we randomly choose one side to go to.
- "see Figure 3 for example of path" -> for an example of a path
- "For the core of the root node it is hold"  (unclear what is meant)

**Questions:**

Q1) I believe the paper would be considerably improved by taking into accound the point W1 stated above. In particular, I suggest the following:
- Briefly recall the rationale and the mechanics of TT-Cross and TTOpt.
- Write Algorithm 1 in a more explicit fashion, clearly stating its inputs and outputs, using a more formal notation.

Q2) In Section 3, the authors state that by HT they "mean a tensor tree that is not necessarily balanced". Thus, the topology of the HT structure used in the numerical experiments should be specified. Is it always taken to be simply a balanced tree?

Q3) Related to the previous question, how is this topology chosen in practice? Is it simply by trial-and-error or are there some heuristics for doing so?

Q4) Intuitively, the higher flexibility of the proposed scheme in comparison with the TT-based methods should lead to smaller approximation errors, provided that the HT structure is correctly chosen. Hence:
- Is it possible to derive any quantitative comparison between the TT and HT approaches?
- More generally, can you give any approximation guarantees of your method for some reasonable classes of functions, say as function of the number of point evaluations?

Q5) Regarding the point W3 raised above, is the novelty here all in the way where HTBB selects the next node that should be visited, the rest of it being similar to TT-Cross and TTOpt?

Q6) Some details and relevant discussion are lacking in the numerical experiments. Namely, what are the TT ranks used in the experiments of sections 6.1 and 6.2? What are the total number of parameters in both models? If the TT ranks are increased (but the HT ones are held fixed), is the TT model able to gradually bridge the performance gap, and if so, at which additional cost?

---

> ### Author Response · Authors · 2024-11-25
>
> Dear Reviewer, thank you very much for your comments, which allowed us to correct the shortcomings and improve the quality of our work. Below are our responses for your comments.
>
> 1. A brief description of the rationale behind the TT-Cross and TTOpt should be given...
>
> We have presented our method in the text, with corresponding references to original papers for TT-Cross and TTOpt approaches. Given the limitations of the text size, it seems difficult to present descriptions of known methods TT-Cross and TTOpt. If you insist on the need for such a description, we can add it to Appendix.
>
> 2. ...is not quite clear what is the output of Algorithm 1 ...
>
> Thank you! We modify the text.
>
> 3. A careful revision is needed ...
>
> Thank you for the typos pointed out, we have corrected them.
>
> 4. Write Algorithm 1 in a more explicit fashion, clearly stating its inputs and outputs, using a more formal notation.
>
> Thank you! We extend its description.
>
> 5. In Section 3, the authors state that by HT they "mean a tensor tree that is not necessarily balanced". Thus, the topology of the HT structure used in the numerical experiments should be specified. Is it always taken to be simply a balanced tree?
>
> Yes, but maybe with unused indices (as in Figure 3, where unused leafs [green boxes] as well as their parents [purple circles] are never visited) since balance tree has exactly $2^n$ elements. It's roughly equivalent to a slightly trimmed tree. However, our algorithm can also support completely unbalanced trees, for example, we can consider a nearly one-dimensional Tucker-TT structure, but we have not yet found the practical advantages of these structures, this is future work.
>
> 6. ... how is this topology chosen in practice?
>
> Just empirically for now.
>
> 7. ... can you give any approximation guarantees of your method ...
>
> We have no theoretical guarantees. But the theorem from in Buczy´nska et al. (2015) states that HT is more expressive.
> Note that the original TT-Cross paper and the TTOpt paper also lacked the convergence or approximation guarantee.
>
> 8. ... is the novelty here all in the way where HTBB selects the next node that should be visited, the rest of it being similar to TT-Cross and TTOpt?
>
> Our main contribution is the ability to obtain an approximation of a multidimensional tensor, such that it does not fit in the memory of a computing system, in the well-known Hierarchical Tucker format. This format was known before us, and various results exist for it, in particular theorems on the expressivity of the format. Thus, this work cannot be considered incremental with respect to the TT-cross algorithm work. In addition, we have combined both approximation and optimization of a black-box function in one algorithm. Our algorithm was inspired by the TT-cross algorithm, we use the same idea of getting the right indices using the MaxVol algorithm. But our difference is not only in the new index traversal procedure. In TT-decomposition each core is as 3-dimensional as in HT. But there each core has a free index that corresponds to the component argument. And when MaxVol is used, the corresponding matrix is unfolded by this index. In the case of HT, all cores dimensions (except leaf cores) are latent and thus symmetric. It is not obvious in this case how to perform unfolding, since the latent core dimensions do not correspond to the argument of the tensor (and black box). In order to work in such a situation, we proposed to associate indices with the connections of cores (nodes) rather than with the cores themselves, we introduced two types of indices - "up" and "down",
> and the possibility of going in any direction (by any connection of a given node with other node). All this is a novelty.
>
> 9. Some details and relevant discussion are lacking in the numerical experiments...
>
> We used the most efficient (in terms of approximation quality) rank-adaptive version of the TT-cross algorithm (i.e. the method automatically adaptively increases the TT-rank what corresponds to the number of optimizable parameters until the budget is exhausted). As noted in the text, for all methods (HTBB and baselines) we set the same limitation on the number of requests to the black box.

---

### Official Review · Reviewer_oAo4 · 2024-11-02

**Soundness:** 3
**Presentation:** 3
**Contribution:** 3
**Rating:** 6
**Confidence:** 4

**Summary:**

In this paper, the authors propose a low-rank hierarchical Tucker cross-approximation method, utilizing the MaxVol index selection procedure. The paper is well-structured, with a comprehensive introduction. In my assessment, this work constitutes a notable contribution to the field.

**Strengths:**

The authors propose a low-rank hierarchical Tucker cross-approximation method, utilizing the MaxVol index selection procedure. The paper is well-structured, with a comprehensive introduction. In my assessment, this work constitutes a notable contribution to the field.

**Weaknesses:**

Please refer to the questions below.

**Questions:**

1. The introduction of the equation in Section 3 appears abrupt. It is recommended that the authors include a figure to provide further illustration and clarity.

2. What would be the performance of Algorithm 2 if QR decomposition were replaced with singular value decomposition (SVD)?

3. Given the existence of a canonical form for the hierarchical Tucker decomposition, could the authors provide a comparison between the cross-approximation method and an SVD-based method in the context of hierarchical Tucker?

4. In the first experiment, in addition to the TT cross-approximation, the Tucker cross-approximation should be considered, as hierarchical Tucker is an extension of the Tucker decomposition.

5. The experimental results should include performance comparison with regard to different tensor orders, as this would provide insight into the scalability of the proposed method.

6. All experimental results should be normalized for comparability.

---

> ### Author Response · Authors · 2024-11-25
>
> Dear Reviewer, thank you very much for your high evaluation of our work and constructive comments! Below we provide our responses.
>
> 1. The introduction of the equation in Section 3 appears abrupt. It is recommended that the authors include a figure to provide further illustration and clarity.
>
> Fig. 3 in the manuscript relates to this equation. We additionally add the following  text in the beginning of Sec. 2: "(see Fig. 3 for a visual example of HT structure)" for the convenience of readers.
>
> 2. What would be the performance of Algorithm 2 if QR decomposition were replaced with singular value decomposition (SVD)?
>
> Just slower. SVD could, theoretically, show more stability, but the results of QR decomposition are simply fed into MaxVol, which itself works with some accuracy, and go nowhere else. So we decided to use QR for acceleration. However, we assume that the black-box computation takes much longer than the internal procedures of our algorithm (and this is exactly what usually happens in practice when approximating or optimizing complex black boxes).
>
> 2. Given the existence of a canonical form for the hierarchical Tucker decomposition, could the authors provide a comparison between the cross-approximation method and an SVD-based method in the context of hierarchical Tucker?
>
> Our goal is to work with tensors that do not fit completely in memory (here we follow the ideology of the TT-Cross algorithm, which can be replaced by a sequential SVD call if the original tensor fits in the memory of the computer system). Of course, if the tensor is small enough and we have the computer memory for SVD-based algorithms, they will be more accurate (and likely faster, for the very small tensors) than ours.
>
> 3. In the first experiment, in addition to the TT cross-approximation, the Tucker cross-approximation should be considered, as hierarchical Tucker is an extension of the Tucker decomposition.
>
> Here we want to note that in our work we consider essentially multidimensional tensors (for example, of dimension 256 or 1024). The Hierarchical Tucker (HT) decomposition, as well as the TT decomposition, are free from the curse of dimensionality, while the ordinary Tucker decomposition contains a kernel with an exponential number of parameters and therefore its use in our case is extremely difficult or impossible.
>
> 4. The experimental results should include performance comparison with regard to different tensor orders, as this would provide insight into the scalability of the proposed method.
>
> We would like to note that within the framework of the problem statement we are considering, it is assumed that the time it takes to calculate the black box can be significantly longer than to perform internal operations of the algorithm and therefore it is logical to measure the efficiency of the algorithm not in its operating time, but in the number of queries made to the black box. As noted in the text, for all methods (HTBB and baselines) we set the same limitation on the number of requests to the black box.
>
> 5. All experimental results should be normalized for comparability.
>
> In the tables we present relative approximation errors and absolute optimization errors (since for many functions the true optimum is at zero).

---

> > ### Comment · Reviewer_oAo4 · 2024-11-26
> >
> > Thank you for your feedback. I will retain the score.

---

### Official Review · Reviewer_qu4L · 2024-11-03

**Soundness:** 2
**Presentation:** 1
**Contribution:** 2
**Rating:** 3
**Confidence:** 4

**Summary:**

This work proposes a method called Hierarchiacl Tucker Black Box (HTBB) based
on low-rank hierarchical Tucker decomposition and the MaxVol subroutine for
finding good submatrices [Goreinov et al., Matrix Methods: Theory, Algorithms
and Applications 2010]. The approach is inspired by the algorithms:
* TT-cross [Oseledets-Eugene, Linear Algebra and Its Applications, 2010]
* TTOpt [Sozykin et al., NeurIPS 2022]

with the goal of extending these works
to hierarchical Tucker tree structures. The main techincal contribution of
this work is Algorithm 2, which decomposes dimensions of the tensor to
construct a good low-rank hierarchical Tucker tree structure. Finally, this
work compares HTBB with several other algorithms across a benchmark of
high-dimensional tensors representing discretized continuous functions.

**Strengths:**

- The main technical details in Section 4 are sophisticated and well organized.
  That said, they could benefit from more context and a better setup in the
  preceeding sections.
- Experiments are run against well defined benchmarks tensors from previous
  work, e.g., [Jamil-Yang, J. of Math. Modeling and Num. Opt. 2013]. These
  tensors are discretized continuous functions with known ground truth.
- Table 3 and Figure 4 in the experiments suggest that HTBB has a lot of
  potential, e.g., there are clear improvements in the $10^4$ number-of-requests
  regime.

**Weaknesses:**

- First and foremost, the presentation quality of this work makes it
  challenging to (1) have a precise understanding of what problem is being
  solved, and (2) the difference between your approach and other existing works.
  The good ideas in this work are getting lost, largely because there is not a
  crisp and standalone mathematical problem definition (relies too much on
  pointing to related work).
- Related works cover many different method types (e.g., particle swarm
  optimization).  How many of these solve the same problem and could be
  compared against in the experiments?

**Questions:**

- For the 14 benchmark instances, [Jamil-Yang, J. of Math. Modeling and Num. Opt. 2013] has 100+
  tensors in their dataset. How did you choose this subset of 14 across multiple works?
- "gradient-free" methods are mentioned regularly throughout the paper. Can you give an example
  of an effective method that you compare against that is gradient-based?
- Are the sizes (i.e., the number of optimizable parameters) in HTBB and TT-cross equivalent in Table 1?
- In Figure 4, why are the SPSA and OPO methods better up until $10^3$ requests?
- Note: The [Goreinov et al., 2010] citation should include the journal.

---

> ### Author Response · Authors · 2024-11-25
>
> Dear reviewer, thank you very much for your analysis of our work and the specific notes you formulated! Below we provide our responses to them. We will be happy to answer any additional questions you may have.
>
> 1. ... what problem is being solved.
>
> As written in Sec. 2 "MOTIVATION AND OVERALL IDEA" (L086--100), our goal is to extend the TT-Cross and TTOpt algorithms to HT tensors. These algorithms perform black-box approximation of a given surrogate model and find the optimum of a black-box function, correspondingly. We write about the same in "our main contributions are the following:" (L068-L072).  We also have separately added a corresponding explanation in the revised version of the manuscript.
>
> 2. ... the difference between your approach and other existing works.
>
> The closest approaches to our one are the TT-Cross and TTOpt algorithms, which work with TT decomposition. We deal with the HT decomposition, and this is our main difference. We write about it in Sec. 2 MOTIVATION AND OVERALL IDEA (L085 - 102).
>
> 3. The good ideas in this work are getting lost ... relies too much on pointing to related work.
>
> We have added a mathematical statement to the revised version of the manuscript. As for references to similar papers, we mostly cite them as motivation and justification for why we expect our algorithm to work well. But our algorithm differs significantly from those already published, and thus we describe its work in enough detail to make its essence clear even to a reader not familiar with previous papers.
> Please clarify what terms or mathematical expressions relating specifically to our algorithm that we have not disclosed that could potentially interfere with understanding how our algorithms work?
>
> 4. Related works cover many different method types (e.g., particle swarm optimization). How many of these solve the same problem and could be compared against in the experiments?
>
> We note that our method combines approximation and optimization, so we compared it separately with an alternative approximation method (TT-cross) and various optimization methods (TTOpt, One+One SPSA, and PSO, i.e., Particle Swarm Optimization).
>
> 5. For the 14 benchmark instances, [Jamil-Yang, J. of Math. Modeling and Num. Opt. 2013] has 100+ tensors in their dataset. How did you choose this subset of 14 across multiple works?
>
> Among those 100+ functions many have the same properties or even just the same functions, but in different benchmarks. Also, many functions are formulated only for a specific dimension (for example, 2). We've made a representative compilation of functions from benchmarks, you've mentioned above based on the following principles: functions should be different, and functions should be hard to optimize using gradient-based methods.
>
> 6. "gradient-free" methods are mentioned regularly throughout the paper. Can you give an example of an effective method that you compare against that is gradient-based?
>
> As we've stated in our research task, the sole focus of our work is on the functions, for which gradient information either cannot be obtained or just is not useful. In both cases comparison with any gradient-based methods makes no sense. For example, in Appendix, we presented the results of additional experiments for four challenging problems of maximization of the cumulative reward for reinforcement learning agents by the proposed method HTBB and various baselines, and gradients are not available for these problems.
>
> 7. Are the sizes (i.e., the number of optimizable parameters) in HTBB and TT-cross equivalent in Table 1?
>
> We used the most efficient (in terms of approximation quality) rank-adaptive version of the TT-cross algorithm (i.e. the method automatically adaptively increases the TT-rank what corresponds to the number of optimizable parameters until the budget is exhausted), so for different benchmarks the ratio in the number of parameters between the two methods turns out to be different.
>
> 8. In Figure 4, why are the SPSA and OPO methods better up until requests?
>
> Both of the methods, you've mentioned, belong to a different family of black-box optimization techniques, so we have neither intuition, nor explanation on the observed behavior. However, even with the limited knowledge, one can notice, that both of the methods have a quite good initial guess and then improve only a little.
>
> 9. The [Goreinov et al., 2010] citation should include the journal.
>
> Thank you for noticing this inaccuracy in the citation. We will correct this oversight.

---

### Official Review · Reviewer_MFtq · 2024-11-03

**Soundness:** 3
**Presentation:** 2
**Contribution:** 2
**Rating:** 3
**Confidence:** 3

**Summary:**

This paper develops new black-box approximation and gradient-free optimization methods for multidimensional functions based on Hierarchical Tucker (HT) decomposition. Numerical experiments were conducted to illustrate the improvement over the prior method that was developed based on Tensor Train (TT) decomposition.

**Strengths:**

- The proposed algorithm achieved better numerical performances than the prior works.
- The authors did a good job explaining the details of the proposed algorithm despite the complicated notations, which are common in tensor-related work.

**Weaknesses:**

- A key concern is the lack of theoretical guarantees. Compared to the existing algorithms like TT-Cross and TTOpt, there is no clear improvement in terms of time complexity, approximation guarantees, or optimization guarantees with respect to the dimension $d$ and rank $r$. The algorithm seems too heuristic to me and the contribution is not enough for an ICLR publication.
- In addition, I have reservations about the practicality of the proposed "black-box" approach. In the numerical experiments, all functions have well-behaved analytical expressions, which may not reflect real-world scenarios. It remains unclear whether the approach would perform well in applications where functions may lack favorable properties, such as continuity or convexity.
- The presentation of the paper, particularly in terms of English language usage, could be improved.

**Questions:**

- typo on page 5, line 47: $i^{down}_{l,j}$
- typo on page 6, line 323: "As each visited node"

---

> ### Author Response · Authors · 2024-11-25
>
> Dear reviewer, Thank you very much for analyzing our work and formulating the specific questions you asked! Below are our responses.
>
> 1. The algorithm seems too heuristic to me and the contribution is not enough for an ICLR publication.
>
> We agree with the reviewer that our algorithm is quite heuristic, and we leave strict guarantees of convergence, accuracy, etc. to future work, presenting in this paper only the essence of the algorithm and numerical results. However, the papers on TT-Cross and TTOpt (which was published on A* NIPS conference) are similar in spirit and do not contain theoretical discussions of aspects of the algorithm. In very general terms, we expect our algorithm to produce better results than the above, since TT-Cross and TTOpt methods work with TT-decomposition, while we operate with HT-decomposition. It has been proved to be more expressive for HT. We write about this on page 2, L085--087. This fact was proved in Buczy´nska et al. (2015). We have extended the corresponding text in the revised version of the paper. Could you please specify to what extent you think theoretical justifications should have been presented and what kind of theorems you would expect to be enough for an ICLR publication?
>
> 2. In the numerical experiments, all functions have well-behaved analytical expressions, which may not reflect real-world scenarios. It remains unclear whether the approach would perform well in applications where functions may lack favorable properties, such as continuity or convexity.
>
> We demonstrate the effectiveness of the proposed approach for 14 multidimensional analytical functions, many of which have complex landscape with several local extremas. This set of functions is often used to test optimization algorithms and surrogate modeling. In Appendix, we also presented the results of additional experiments for four challenging problems of maximization of the cumulative reward for reinforcement learning agents by the proposed method HTBB and various baselines. Could you please clarify which BB you would recommend for further validation of our algorithm?
>
> 3. The presentation of the paper, particularly in terms of English language usage, could be improved.
>
> Thank you, we have proofread the text of the manucript and tried to make improvements to the presentation.
>
> 4. Questions.
>
> Thank you for the typos pointed out, we have corrected them.

---

> > ### Comment · Reviewer_MFtq · 2024-11-26
> >
> > The examples of theoretical justifications include the approximation error of the algorithm, the iteration complexity to achieve error $\epsilon$, or the time complexity. I believe that presenting only heuristic algorithms falls short of the standards typically expected at top machine learning conferences, particularly in the area of optimization, unless they demonstrate significant improvements in real-world applications. Therefore, I retain my original score.

---

### Meta-Review · Area_Chair_1WWS · 2024-12-23

**Metareview:**

This work proposes to approximate a black-box function of discrete inputs with hierarchical Tucker decomposition. A new heuristic algorithm is presented, but no theoretical guarantee is provided. Numerical experiments are conducted on some well-behaving functions, therefore not particularly convincing about its universal approximation ability.

Readability could be improved.

**Additional Comments On Reviewer Discussion:**

The authors provided rebuttal to all the reviews, but are not convincing enough.

---

### Decision · Program_Chairs · 2025-01-22

Reject